# How does symbolic success affect redistribution in left-wing voters? A focus on the 2017 French presidential election

**Vincent Berthet** [1,2]*, **Camille Dorin**[2], **Jean-Christophe Vergnaud**[2], **Vincent de Gardelle**[3]

**1** Psychology and Neuroscience Lab, EA 7489, Université de Lorraine, Nancy, France, **2** Centre d'Economie de la Sorbonne, CNRS UMR 8174, Paris, France, **3** CNRS and Paris School of Economics, Paris, France

* vincent.berthet@univ-lorraine.fr

**Data Availability Statement:** All data files are available from the Mendeley database (DOI: 10. 17632/nkx7z2zfmn.2).

## Abstract

Redistribution preferences depend on factors such as self-interest and political views. Recently, Deffains et al. (2016) reported that redistributive behavior is also sensitive to the actual experience of success or failure in a real effort task. While successful participants ('overachievers') are more likely to attribute their success to their effort rather than luck and opt for less redistribution, unsuccessful participants ('underachievers') tend to attribute their failure to external factors and opt for more redistribution. The aim of the present study was to test how the experience of success (symbolic success) and political views interact in producing redistributive behavior in an experimental setting. The study was conducted during the 2017 French presidential election. Our sample was biased towards left-wing, and most participants reported voting for Mélenchon, Hamon or Macron. Our findings reveal that 1) Macron voters redistribute less than Hamon voters who themselves redistribute less than Mélenchon voters, 2) overachievers redistribute less than underachievers only among Mélenchon voters. This suggests that redistributive behavior is governed primarily by political opinions, and that influence by exogenous manipulation of symbolic success is not homogenous across left-wing political groups.

## Introduction

Support for redistribution varies greatly across individuals within a society, and is a major component of their political positioning. Political parties put forward different redistributive policies in their respective agendas. Accordingly, understanding the determinants of support for redistribution has been a topic of major interest for researchers in economics and political sciences.

One can distinguish two main factors contributing to this support, namely self-interest and fairness considerations [1]. On the one hand, the individual attitude towards a more redistributive or a less redistributive system is shaped by the economic self-interest of the individual, i.e. the effect that the redistributive system has upon the individual's net income. Obviously, self-interest pushes wealthy individuals to support redistribution less than poor individuals. On the other hand, support for redistribution is also dependent on fairness considerations [2,3]. The redistributive policy chosen in a society reflects the beliefs about the determinants of income inequality and the main causes of poverty [1]. If wealth is primarily determined by

**Funding:** The author(s) received no specific funding for this work.

**Competing interests:** The authors have declared that no competing interests exist.

chance or by factors that are not under the control of individuals, then support for redistribution increases [4,5], in accordance with the accountability principle [6].

Surveys have shown that such beliefs about the determinants of inequality are not homogeneous across the population [e.g. 7]. Relatedly, support for redistributive policies varies across social groups defined by race, gender, age or socioeconomic status [8]. In the United States, whites are more averse to redistribution than blacks, even after controlling for individual characteristics such as income, education, etc. [e.g., 9,10]. Past upward mobility also decreases the support for redistribution [e.g., 10,11]. Some of these observations have been confirmed by experimental data. For instance, when participants are presented with mock news articles reporting high (vs. low) rates of social mobility, their tolerance for inequality increased [12]. Providing American adults with factual information about the rise of inequalities in the United States (vs. control information) increased their beliefs that economic inequalities are due to structural rather than individual factors and increased support for redistribution [13,14].

The present work follows up on a recent study by Deffains, Espinosa, and Thöni [15] who introduced an exogenous manipulation of status and found this manipulation to affect the redistributive behavior of participants, even when self-interest was not at stake. After a real effort task, each subject was randomly given a status of either 'overachiever' (performance above the median) or 'underachiever' (performance below the median). In a subsequent disinterested dictator game, participants were asked to reallocate money between two randomly chosen individuals in their session, from the richest to the poorest individual. It turned out that on average, overachievers redistribute less than underachievers. The information provided to the subjects about the determinants of task performance (i.e. luck or effort) was very vague, and the authors found that overachievers also emphasized more the role of effort in their outcome than underachievers. Noteworthy, Deffains et al. suggested that participants exhibit a self-serving bias [16] by adopting beliefs favorable to them. More precisely, successful individuals attribute their own success to effort and others' failure to a lack of effort, and in accordance with the accountability principle, they believe that no redistribution should take place. On the contrary, unsuccessful individuals attribute their own failure to bad luck and others' success to favorable circumstances, so they support redistribution towards the most disadvantaged.

Since beliefs about the role of luck can be affected both by exogenous manipulations [15] and political opinions [e.g., 17], one could anticipate that these two factors may interact in their influence on redistributive behavior. The goal of the present study is to evaluate this interaction. To do so, we tested an exogenous manipulation of status much like Deffains et al., while evaluating political opinions of participants, in the context of the French 2017 presidential election.

Is the effect of status uniform across the different voters? More precisely, we hypothesized that the exogenous manipulation of Deffains et al. would have an effect on redistributive behavior for subjects who hold moderate political views but no effect for subjects who hold extreme political views. The 2017 French presidential election provided a unique opportunity to compare extreme voters to moderates. Indeed, in 2017, most electors moved away from the candidates of the two major traditional parties (Hamon for the left-wing "Parti Socialiste" vs. Fillon for the right-wing party "Les Républicains"), who together gathered only 25% of the votes in the first round. Instead, electors supported the moderate candidate Macron (who eventually won the election) and the candidates of radical parties (Mélenchon for the far-left and Le Pen for the far-right). In other words, as was seen in other western democracies in the last decade, this election moved away from the traditional left-right opposition towards a center-extreme polarization.

## Method

### Participants

A total of 649 unpaid participants completed the experiment (see "Description of our sample" below). Participants were essentially French people who responded to an announcement we posted on the Parisian Experimental Economics Laboratory (LEEP) portal, the Paris School of Economics portal, and the main social networks (Facebook and Twitter) inviting them to participate in an online survey on the presidential election. The website that hosted the experiment provided participants with all information about the research (the purpose and nature of the study, the voluntary nature of participation, and the possibility of withdrawing from the experiment at any time without any penalty or consequences). This research was reviewed and approved by Institutional Review Board–Ecole d'économie de Paris (approval number: IRB00010601). Written informed consent was obtained from all participants.

### Procedure and measures

The experiment took place during the two weeks separating the two rounds of voting in the 2017 French presidential election (April 23-May 7). Participants first performed a computerized effort task without monetary reward linked to performance. This task was an Implicit Association Test (IAT) aimed at measuring their implicit attitude towards France (in its preliminary version, this study was intended to examine to what extent implicit and explicit attitudes predict participants' voting intention. Because of our skewed sample, however, we could not really evaluate properly the voting intention towards Marine Le Pen. Thus, the variable "voting intention for the second round" was not taken into account in the analysis. We then focused our analysis essentially on the determinants of redistributive behavior). Participants were asked to respond as fast and accurately as possible, and they were informed that their performance would be their mean reaction time over the task. After completion of the task, participants were given a (fake) feedback on their performance and were randomly assigned to the overachiever or underachiever groups (status). Then, they completed a disinterested dictator game in which they were asked to reallocate money between two fictive individuals, a rich and a poor individual. The game was scripted as follows: "Imagine that 100 euros were allocated to two participants A and B based on their performance on the previous speeded-response task. A received 80 euros based on her good performance, B received 20 euros based on her weak performance. If you could reallocate the 100 euros to A and B, how would you reallocate them?" Participants chose the amount of money (between 50 and 100) they would allocate to A, B receiving the rest. Next, participants responded to five self-report items on a 7 points Likert scale measuring fatalism ("to what extent do you relate you performance to 1: chance or 7: effort); their views on income inequality (1: egalitarian, 7: liberal); their attitudes towards economic patriotism ("Do you think that the French government should take more patriotic measures in the economic and the social domain?", 1: unfavorable, 7: favorable); their attitudes towards France ("Do you like France?" 1: positive, 7: negative); and their political position on the left-right continuum (1: extreme left, 7: extreme right). Then, participants reported their vote in the first round. Here, the response modalities included the 11 candidates involved plus the two options "I did not vote in the first round" and "I voted blank or null in the first round". Finally, participants reported their voting intention for the second round. At this stage, four response modalities were presented: "I will vote for Macron", "I will vote for Le Pen", "I will vote blank or null", and "I will not vote".

## Results

### Description of our sample

Participants were 357 females and 292 males (mean age 33.62 years, *SD* = 15.44 years) (due to a technical error in the data collection, redistribution choices could only be analyzed for 626 participants). Regarding the socio-professional category, it turned out that managers and white-collar professions (34.36%) and students (41.60%) were overrepresented in our sample (both categories representing 75% of the sample). Regarding reported votes for the first round, our sample was clearly left-wing oriented, and voters for the two main right-wing candidates (Fillon and Le Pen) were underrepresented, whereas voters for Mélenchon, Hamon, and Macron were overrepresented (Fig 1). Therefore, in subsequent analyses we focus on participants who reported having voted for Mélenchon, Hamon, or Macron in the first round of the election (*N* = 506, 78% of the initial sample), given the lack of data for the other cases. Accordingly, in what follows the variable "First-round vote" is a categorical variable with 3 possible values, namely Mélenchon, Hamon, and Macron. Table 1 reports the age, gender, socio-economic category, and average status for the different group of voters in our sample.

As this selection resulted in a restriction of variance of the Political position variable (Fig 2), we considered the vote reported for the first round (hereafter First-round vote) as the only measure of political opinions in the analysis. Note that at the time of this experiment, Mélenchon, Hamon, and Macron were all considered left-wing candidates. Specifically, Mélenchon was considered as the main candidate of the radical left, Hamon was the official candidate of the major French left-wing party ("Parti Socialiste"), and Macron was associated both with a left-wing government under former president Hollande and with a social-liberal position with a pronounced liberal component.

### Status manipulation

Table 2 reports age, gender, socio-economic category and First-round vote of overachievers and underachievers, showing that our random manipulation of status did not create an unwanted bias between overachievers and underachievers.

### Redistributive behavior and self-report measures

Table 3 indicates the descriptive statistics (means and standard deviations) and the correlations between the different behavioral and personality measures. We note that almost all pairwise correlations measures were significant, except for the correlation between Fatalism and Political position. In particular, the share given to the richer player in the disinterested dictator game, which quantifies participants' attitude towards income inequality in a simple behavioral test, was correlated positively with the explicit attitude towards inequality ($r = 0.24$, $p < 0.001$), which was most correlated with the stated political position ($r = 0.60$, $p < 0.001$).

### Effect of status on redistribution

The main point of interest of the analysis was how the amount of money (between 50 and 100) reallocated to the "richer agent" A in the disinterested dictator game was affected by the manipulation of Status and by the vote reported by participants. To assess this, we conducted a 2 (First-round vote) × 2 (Status) ANOVA for independent samples on the share given to A as a dependent variable. This analysis yielded a significant main effect of First-round vote ($F(1, 500) = 8.65$, $p = .0002$, $\eta_\text{p}^2 = 0.0334$), with no main effect of Status ($F(1, 500) = 1.72$, $p = .19$, $\eta_\text{p}^2 = .0034$), but an interaction between First-round vote and Status ($F(2, 500) = 3.16$, $p = .043$, $\eta_\text{p}^2 = .0124$). The main effect of First-round vote confirmed our expectations that participants who reported voting

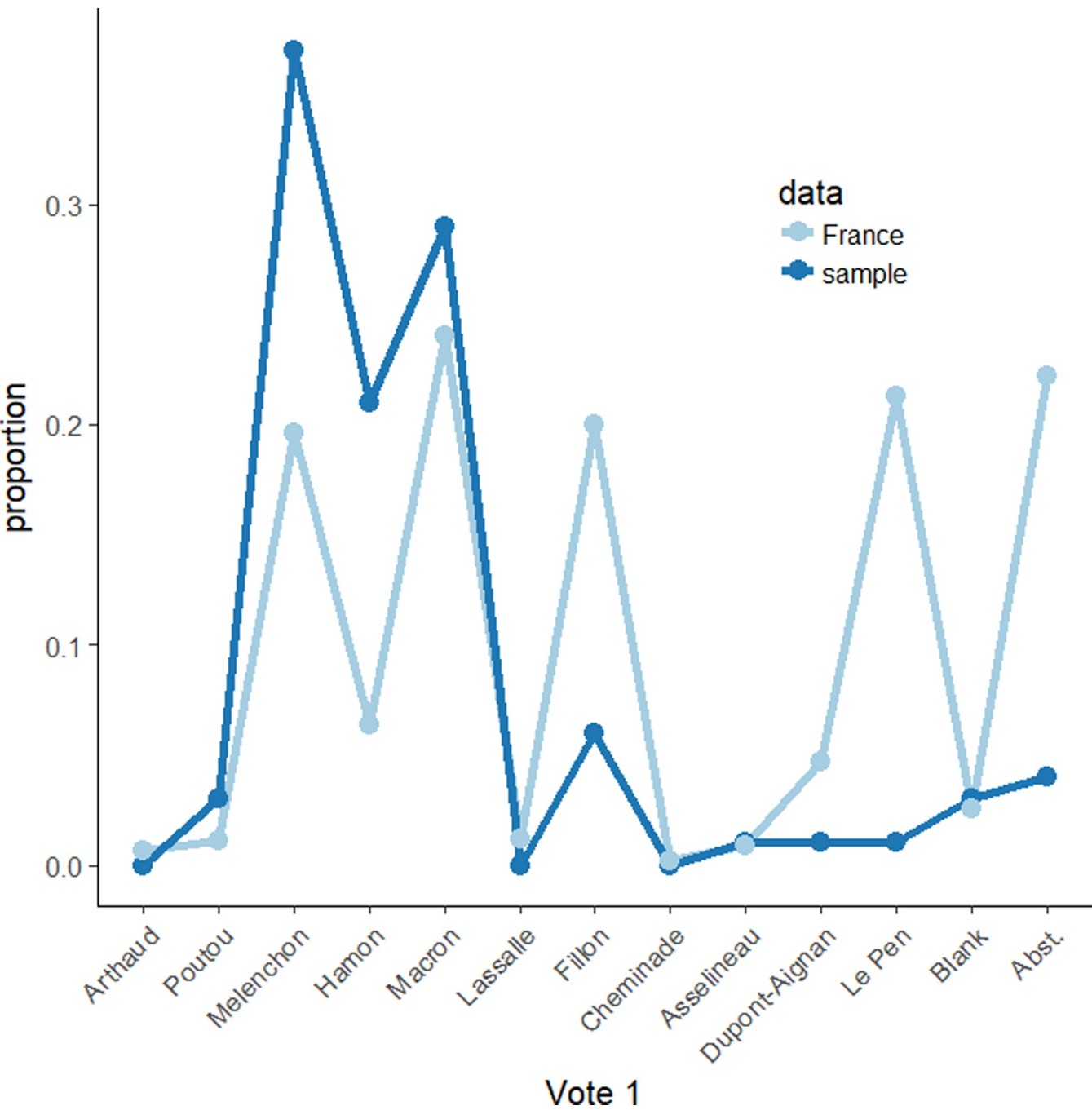

**Fig 1. Distribution of the First-round vote and comparison with actual results at the national level.**

for more left-wing candidates would also exhibit greater redistribution. Indeed, Mélenchon voters allocated a mean amount of 58.45 to A ($SD = 11.50$), Hamon voters 60.54 ($SD = 12.04$), and Macron voters 63.65 ($SD = 13.18$). The manipulation of Status did not produce a statistically significant effect, although the observed pattern was in the expected direction. Indeed, overachievers allocated slightly more to A ($M = 61.39$, $SD = 12.81$) than underachievers ($M = 59.95$, $SD = 11.94$). This is comparable to what was reported by Deffains, et al. [15] (converted to our

**Table 1. Socio-demographic characteristics of participants reporting voting for Mélenchon, Hamon or Macron in the first round of the election in our dataset.** Participants who reported another vote are pooled together in this table, and were not analyzed further in the present study.

| First-round vote | N | Age (SD) | Gender (% women) | Occupation (% White Collar) | Occupation (% Student) | Status (% overachiever) |
|---|---|---|---|---|---|---|
| Mélenchon | 219 | 31.21 (12.42) | 0.62 | 0.29 | 0.45 | 0.46 |
| Hamon | 121 | 32.57 (14.59) | 0.67 | 0.33 | 0.45 | 0.57 |
| Macron | 166 | 37.22 (16.75) | 0.45 | 0.47 | 0.32 | 0.47 |
| Other | 120 | 32.40 (14.65) | 0.46 | 0.23 | 0.46 | 0.43 |

measure, they respectively obtained M = 61.46, SD = 14.8 for overachievers and M = 59.95, SD = 11.94 for underachievers). With regards to the interaction, we expected that Status would have an effect on redistributive behavior for participants with moderate political views (Hamon and Macron voters), but not necessarily for those with extreme political views (Mélenchon voters), whom we expected to strongly redistribute irrespectively of our Status manipulation. Inspection of the different groups (Table 4) however revealed a different pattern, and separate analyses for each group of voters indicated that the effect of Status was significant for Mélenchon voters (*F*

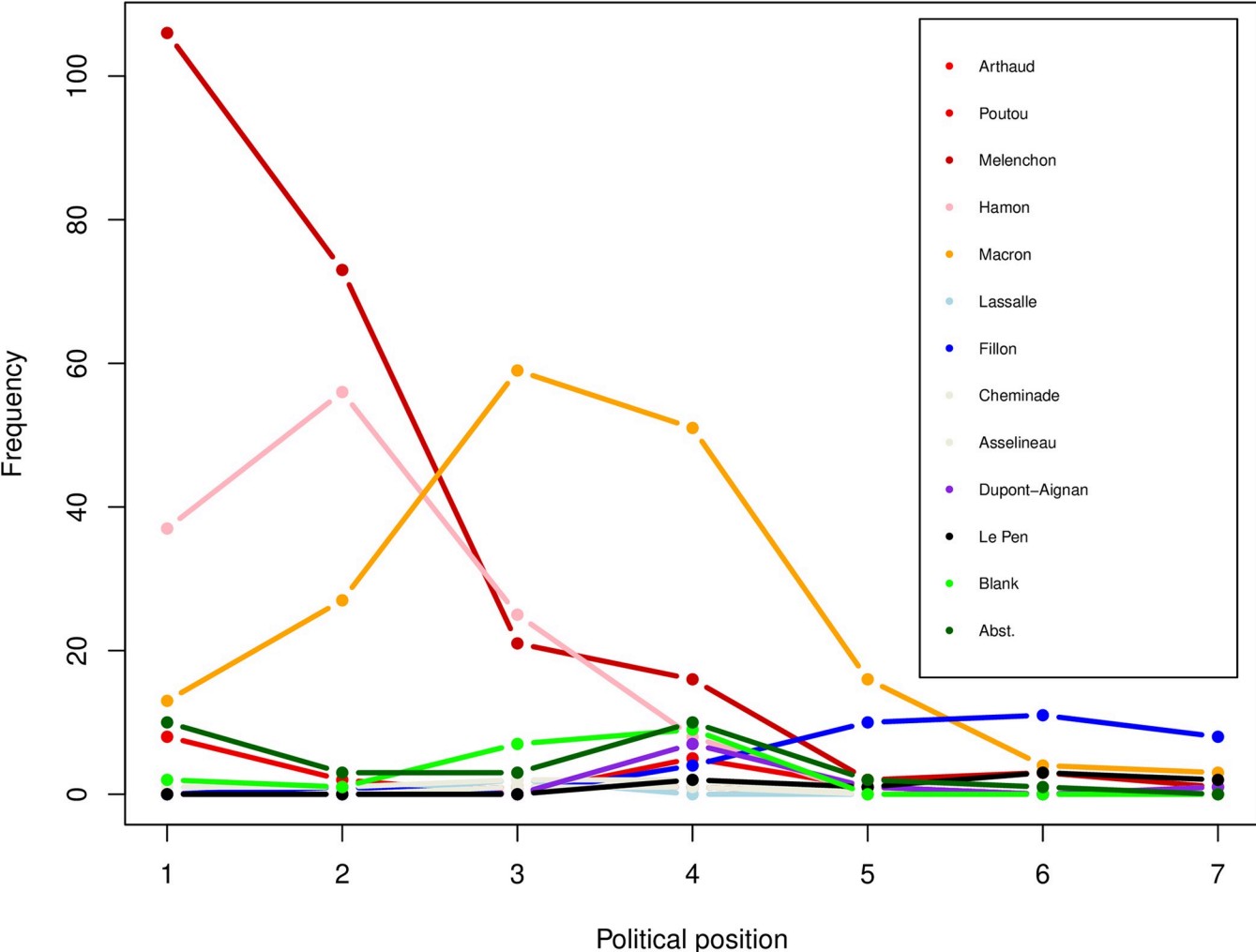

**Fig 2. Relation between First-round vote and political position (1: extreme left, 7: extreme right).**

**Table 2. Comparison of First-round vote and the socio-demographic characteristics of participants in the overachiever and underachiever conditions in our final sample (i.e. including only participants who reported voting for Mélenchon, Hamon or Macron).**

| Condition | N | Age (SD) | Gender (% women) | First-round vote Mélenchon / Hamon / Macron |
|---|---|---|---|---|
| Overachiever | 248 | 33.73 (15.34) | 0.54 | 0.41 / 0.28 / 0.31 |
| Underachiever | 258 | 33.28 (14.06) | 0.60 | 0.46 / 0.20 / 0.34 |
| test | | t = 0.344 | $\chi^2$ = 1.64 | $\chi^2$ = 1.10 / 3.67 / 0.29 |
| p | | 0.731 | 0.2 | 0.29 / 0.06 / 0.59 |

(1, 217) = 6.30, $p$ = .0128, $\eta_p^2$ = .0282), but not for Hamon voters ($F$(1, 119) = 1.99, $p$ = .16, $\eta_p^2$ = .0164) or Macron voters ($F$(1, 164) = 0.50, $p$ = .47, $\eta_p^2$ = .0030) . In other words, we found the opposite pattern to that expected.

Noteworthy, to evaluate whether our results were robust to changes in model specification, we conducted a new regression analysis, in which we added gender and age as covariates (Table 5). This regression revealed that gender affected redistribution, with women redistributing more than men, replicating previous findings [e.g., 2,18]. This analysis also indicated a main effect of the First-round vote, and confirmed the interaction between Status and First-round vote. When examining the effect of Status separately for the 3 groups of voters, again adding gender and age as covariates, we found that redistributive behavior was affected by Status only for Mélenchon voters (F(1, 215) = 5.54, p = .020, $\eta_p^2$ = .0251), but not for Hamon voters (F(1, 117) = 2.09, p = .151, $\eta_p^2$ = .0176) or Macron voters (F(1, 162) = 0.454, p = .502, $\eta_p^2$ = .0028) replicating our main finding. For completeness, we also report in S1 Appendix the results of a regression over all participants, including those who reported a different First-round vote.

Finally, we evaluated the effect of Status on the fatalism measure, that is, the extent to which participants related their performance to chance or effort. We found no evidence that fatalism was affected by Status, $F$(1, 504) = 0.40, $NS$. In other words, we found no evidence for a self-serving bias in our participants, unlike Deffains et al. [15]. Note that the task used in the present study was an IAT, which was objectively less effortful than the counting task used by Deffains et al. Therefore, unlike their participants, participants in our study might have not believed that their performance could be impacted by the amount of effort deployed.

## Discussion

The present study capitalized on a major political election (the 2017 French presidential election) in order to investigate how redistributive behavior is affected by political views and–

**Table 3. Descriptive statistics and zero-order correlations between individual measures: The share left to the richer agent in a disinterested dictator game, fatalism (relating performance to 1: chance 7: effort), income equality (from 1: egalitarian to 7: liberal), economic patriotism (from 1: unfavorable to 7: favorable), attitude towards France (from 1: negative to 7: positive), and political position (from 1: extreme left to 7: extreme right) (N = 649).**

| | | Mean | SD | 1 | 2 | 3 | 4 | 5 | 6 |
|---|---|---|---|---|---|---|---|---|---|
| 1 | Disinterested dictator | 60.08 | 15.19 | – | | | | | |
| 2 | Fatalism | 4.78 | 1.51 | .17*** | – | | | | |
| 3 | Income inequality | 2.98 | 1.78 | .24*** | .10** | – | | | |
| 4 | Economic patriotism | 3.40 | 1.91 | .16*** | .09* | .20*** | – | | |
| 5 | Attitude France | 5.72 | 1.34 | .14*** | .19*** | .20*** | .16*** | – | |
| 6 | Political position | 2.68 | 1.53 | .22*** | .04 | .60*** | .27*** | .19*** | – |

*$p < .05$,

**$p < .01$,

***$p < .001$, two-tailed.

**Table 4. Means and standard errors of the amount of money allocated to A in the disinterested dictator game as a function of First-round vote and status.**

| First-round vote Status | Mélenchon | Hamon | Macron |
|---|---|---|---|
| Overachiever | 60.5 (1.28) | 59.2 (1.30) | 64.4 (1.58) |
| Underachiever | 56.7 (0.91) | 62.3 (1.86) | 63.0 (1.33) |

experimentally induced–symbolic success [15]. We found an overall effect of First-round vote on redistribution such that the mean amount redistributed by the three main groups of voters in our sample was coherent with their respective positions on the left-right continuum. While participants who reported voting for Mélenchon (presumably the most leftists) were the most redistributive, Macron voters (the most liberal) were less redistributive, with Hamon voters falling in between. This finding confirms previous research reporting that preferences for redistribution and progressive taxation are coherent with vote choice: during the French 2012 presidential elections, strong supporters of redistribution voted for the left-wing candidate Hollande, while supporters of a flat rate tax voted for the right-wing candidate Sarkozy [11].

Our main result is that redistributive behavior is influenced by the exogenous manipulation of Status only in a subgroup of participants, specifically those who reported voting for Mélenchon. Therefore, our study partially replicated the findings of Deffains and colleagues [15]. This partial discrepancy between our study and that of Deffains might be due to incentives. In Deffains' study, participants' redistribution choices in the dictator game had real consequences on the payoffs of other players, whereas in our paradigm redistribution choices were only hypothetical. It is possible that incentives might have influenced our results independently of the desirability bias. Participants who reported voting for Hamon or Macron might more sensitive to the presence of real life incentives than Mélenchon voters. Thus, incentivizing redistribution choices might be a necessary feature to obtain the effect of Status in Hamon or Macron voters, whereas Mélenchon voters would exhibit the effect of Status even in the absence of incentives. To evaluate these possibilities, further research would need to compare redistribution choices with and without incentives, for the different groups of voters.

It has been proposed [e.g., 19] that in the absence of incentives, participants might try to please the experimenter or conform to some social norms, e.g. by being generous in dictator games. Could this desirability bias explain our results or the difference between our study and Deffains' study? We believe that such an explanation is unlikely for several reasons. First, if a desirability bias was more present our study than in Deffains' study, then we should have observed more redistribution in our participants. However, in our experiment, participants redistributed less than in Deffains' study: our mean allocation to A was 60.08 while the corresponding value in Deffains' study would be 57.56. Second, and more generally, it is not clear to us why this desirability bias would lead to the specific interaction between Status and First-

**Table 5. ANOVA table for redistributive behavior in the disinterested dictator game.** The different factors included in the model are the effects of gender, age, status in the experiment (overachiever vs. underachiever), First-round vote and the interaction between status and First-round vote.

| | $\eta_p^2$ | S.S. | d.f. | F | p |
|---|---|---|---|---|---|
| Gender | 0.0080 | 584 | 1 | 4.001 | .046 |
| Age | 0.0051 | 373 | 1 | 2.553 | .111 |
| Status | 0.0026 | 188 | 1 | 1.291 | .256 |
| Vote1 | 0.0239 | 1781 | 2 | 6.101 | .002 |
| Status:Vote1 | 0.0119 | 874 | 2 | 2.992 | .051 |
| Residuals | | 72701 | 498 | | |

round vote. Third, the instructions given to participants (see S2 Appendix) did not refer to the aim of our experiment, so participants were naïve about our hypothesis. Had they tried to guess our expectations, we would have found an effect of status on fatalism, which we did not observe either in the full sample (p = .52) nor in Mélenchon voters (p = .35), whose redistributive behavior was affected by status however. Finally, our experiment was conducted online and responses were anonymous, so participants have no pressure to please the experimenter or conform to social norms.

Our study provided a nuanced picture of how redistributive behavior is jointly influenced by political views and the actual experience of individuals (here, the experience of success or failure in a simple decision task). In fact, we hypothesized that the exogenous manipulation of Status would have an effect on redistributive behavior for subjects who hold moderate political views (Hamon or Macron voters), but no effect for subjects who hold extreme political views (Mélenchon voters) who would be more likely to resist any experimental manipulation. Our findings revealed a significant interaction between Status and First-round vote but the pattern we found is the opposite of our expectation, as the only group of voters who were significantly affected by status were Mélenchon voters. Being the most left-wing voters in our sample, endorsing pronounced egalitarian views of society, these voters were supposed to be the most redistributive overall (which was actually observed) but also the least sensitive to the information regarding Status (which was the opposite of what we observed). That result is even more surprising since they reported the most egalitarian views on income ($M = 2.07$) compared to Hamon voters ($M = 2.60$) and Macron voters ($M = 3.86$), $F(2, 503) = 69$, $p < 0.001$. Explanations of this finding in terms of age, sex, or socio-economic status are unlikely in our dataset as Mélenchon voters and Hamon voters did not differ significantly on these variables (Table 1). In addition, we verified that our Status manipulation was truly random with respect to age, sex, or socio-economic status, which did not differ between overachievers and underachievers (Table 2).

Here, we suggest one explanation for our finding that Mélenchon voters were the most affected by Status manipulation. It is worth noting that these voters were also the most versatile at the end of the electoral campaign. Indeed, the dynamics of voting intentions as measured by the polls during the month preceding the first round revealed that voting intentions for Macron remained stable around 23%, those for Hamon collapsed from 12% to 6%, while those for Mélenchon jumped from 11% to 18%. As a candidate, Mélenchon also used a communication strategy based on social influence, with a strong presence on social media, and a populist attitude that emphasized the proximity to his base ("the people"). Individuals that are highly susceptible to social influence were then more likely to become Mélenchon voters, and in our study they were also more likely to be influenced by the Status manipulation. Thus, our result could be explained by susceptibility to social influence as a common cause of voting behavior and of the effect of the Status manipulation.

Before concluding, we must acknowledge several important limitations of our study. First, our sample was limited and was not representative of the French population. In particular, our data could not allow us to investigate the sensitivity of redistribution behavior to an experimental manipulation of success for right-wing voters. One main reason for this limitation was probably the recruitment procedure employed, which was based on social media, local networks, and word of mouth. As a result, according to their reports, our participants were mostly young, left-wing supporters, and closely related to the academic sector. In particular, we did not have enough right-wing or far-right supporters to perform meaningful analyses on this part of the political spectrum. By contrast, analyses of Twitter activity at that period revealed the emergence of three main communities in the French political environment, namely supporters of Macron and Hamon, supporters of Le Pen, and supporters of Mélenchon [20]. It is

possible that redistributive behavior and its sensitivity to our manipulation would have been different for right-wing and far-right voters.

We note that although our results may not be representative of right-wing voters, one could envision that they would generalize for left-wing voters in some other countries. Indeed, in the last decade, western democracies have seen a polarization of opinions, with a crisis of the traditional parties and a rise of support for extreme populist parties. Examples of populist far-left parties are die Linke in Germany, Podemos in Spain, Siriza in Greece, La France Insoumise (Jean-Luc Mélenchon's party) in France. According to Rooduijn and Akkerman [21] these radical left parties have in common that "They do not focus on the 'proletariat', but glorify a more general category: the 'good people'" , contrary to former communist parties and that"- they do not reject the system of liberal democracy as such, but only criticize the political and/ or economic elites within that system". Our results regarding the susceptibility of Mélenchon voters to status manipulation could thus be evaluated and replicated in other countries.

In addition, one could argue that a second limitation of the present work is related to the specific timing of the study, which took place during the French presidential election. This specific timing was chosen on purpose for two reasons. One reason was to benefit from the increased interest towards political topics at this time. The other reason was to probe voters' redistributive behavior at a time that constitutes an important step in the democratic process. However, we acknowledge that it is possible that voters' behavior in our study is unusual, because of this unusual timing. Voters may receive more information in the context of an election, and they may react more strongly to information delivered in this context. Whether our results would generalize to another context unrelated to a particular election thus remains an open empirical issue.

The third limitation relates to the possible discrepancy between actual votes and reported votes in our participants. Poll estimates (based on self-reported votes) and actual votes can indeed differ, as famously illustrated in the 2016 US presidential election, the 2016 "Brexit" referendum, or the 2002 French presidential election, amongst others. However, we note that in the case of the election under study here (2017 French presidential election) the last polls were very accurate. One reason for the discrepancy between self-reported votes and actual votes might be a social desirability bias by which right-wing or far-right votes are expressed less easily and therefore under-estimated in opinion polls [see e.g. 22]. Critically, polling institutes use adjustment procedures to take into this bias when producing their estimates, but we did not. Therefore, right-wing opinions/votes in our sample might have been under-estimated. In sum, although we followed the common practice in studies of voting behavior, and used the terms "Mélenchon voters", "Hamon voters" or "Macron voters", one should bear in mind that our data is about self-reported votes, which might have differed from actual votes.

To conclude, our findings revealed that self-reported far-left voters turned out to be the more sensitive to the exogenous manipulation of symbolic success. This leads to three remarks. Firstly, we need further research to better understand to what extent, and in which groups, redistributive behavior can be manipulated through exogenous manipulations of experience of success. In particular, further studies are needed that shall use a proper manipulation of symbolic success and representative samples in terms of political and socio-economic features. Secondly, our findings suggest that the various political groups process information differently, that is, they are not cognitively homogeneous [e.g., 23–25]. Finally, and more broadly, the fact that Mélenchon voters displayed a different behavior than Hamon and Macron voters extends recent findings showing that supporters of extreme political groups have different characteristics from those with more moderate views, although they are not necessarily different on socio-demographic variables such as age or level of education [e.g. 26]. For instance, Hanel, Zarzeczna, and Haddock [27] reported that extreme (left-wing or right-wing) supporters are

usually more heterogeneous than moderate ones in terms of human values and politics-related variables such as attitudes toward immigrants and trust in institutions. In the current social and political context, we believe that understanding further these differences, especially whether some groups are more susceptible to influence than others, appears a worthwhile subject for future research. Using controlled experiments during political elections can be a useful tool in such research.

## Supporting information

**S1 Appendix. ANOVA table for redistributive behavior in the disinterested dictator game including all groups of voters.**
(DOCX)

**S2 Appendix. Instructions (screen shot and translation).**
(DOCX)

## Author Contributions

**Conceptualization:** Vincent Berthet, Camille Dorin, Jean-Christophe Vergnaud, Vincent de Gardelle.

**Data curation:** Vincent de Gardelle.

**Investigation:** Vincent Berthet.

**Methodology:** Vincent Berthet, Camille Dorin, Jean-Christophe Vergnaud, Vincent de Gardelle.

**Resources:** Jean-Christophe Vergnaud, Vincent de Gardelle.

**Software:** Vincent Berthet.

**Writing – original draft:** Vincent Berthet, Vincent de Gardelle.

**Writing – review & editing:** Vincent Berthet, Vincent de Gardelle.

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
