## [Decision Letter · Decision Letter 0]

16 Oct 2019

PONE-D-19-26429

How symbolic success affects redistributive behavior? An experiment based on the 2017 French presidential election

PLOS ONE

Dear Dr. Berthet,

Thank you for submitting your manuscript to PLOS ONE. After careful consideration, we feel that it has merit but does not fully meet PLOS ONE’s publication criteria as it currently stands. Therefore, we invite you to submit a revised version of the manuscript that addresses the points raised during the review process.

Please find below the reviewer's comments, as well as those from my own.

We would appreciate receiving your revised manuscript by Nov 30 2019 11:59PM. To enhance the reproducibility of your results, we recommend that if applicable you deposit your laboratory protocols in protocols.io, where a protocol can be assigned its own identifier (DOI) such that it can be cited independently in the future. For instructions see: http://journals.plos.org/plosone/s/submission-guidelines#loc-laboratory-protocols

We look forward to receiving your revised manuscript.

Kind regards,

Valerio Capraro

Academic Editor

PLOS ONE

Journal Requirements:

Additional Editor Comments (if provided):

I have now collected one review from one expert in the field. Unfortunately, I was unable to find a second reviewer. However, since this review is very detailed and thorough, I have opted for making a decision on your manuscript with only one review. As you will see, the reviewer is quite positive, but has several major comments. Therefore, I would like to invite you to revise your manuscript according to the reviewer's comment. Additionally, after reading your manuscript, I would like to add one more comment. I have myself done a lot of work on redistributive behavior, which I think is quite relevant to your discussion regarding the determinants of redistributive behavior, so you might want to have a look at it (of course, citing this work is not a requirement). In Capraro & Rand (2018) and Tappin & Capraro (2018) we found that redistributive behavior in the trade-off game is driven by moral preferences for doing the right thing and, consequently, it very much depends on how the decision problem is framed; in Capraro (2019), I found that women are more likely than men to choose an equitable distribution over an efficient one.

Looking forward for the revision.

References

Capraro V (2019) Gender differences in the equity-efficiency trade-off. Available at https://ssrn.com/abstract=3386124.

Capraro V, Rand DG (2018) Do the right thing: Experimental evidence that preferences for moral behavior, rather than equity and efficiency per se, drive human prosociality. Judgment and Decision Making 13, 99-111.

Tappin BM, Capraro V (2018) Doing good vs. avoiding bad in prosocial choice: A refined test and extension of the morality preference hypothesis. Journal of Experimental Social Psychology 79, 64-70.

Reviewers' comments:

Reviewer's Responses to Questions

**Comments to the Author**

1. Is the manuscript technically sound, and do the data support the conclusions?

Reviewer #1: Partly

2. Has the statistical analysis been performed appropriately and rigorously? 

Reviewer #1: No

3. Have the authors made all data underlying the findings in their manuscript fully available?

Reviewer #1: Yes

4. Is the manuscript presented in an intelligible fashion and written in standard English?

Reviewer #1: Yes

5. Review Comments to the Author

Reviewer #1: Referee’s report - Manuscript ID PONE-D-19-26429

Title: " How symbolic success affects redistributive behavior? An experiment based on the 2017 French presidential election" for PLOS ONE

This paper uses experimental data to examine how redistributive preferences are shaped by the experience of success and political views. This experiment was implemented during the recent presidential election in France. It finds that redistributive behavior is affected primarily by political opinions e.g. Mélenchon voters redistribute more than other groups. In addition, the paper shows that redistributive behavior is influenced by the exogenous manipulation of experiences of success. Interestingly, this effect is not homogenous across political groups and only the Mélenchon voters were significantly affected by the status.

The paper is well-written and competently conducted. The literature review is fine although I would suggest referring also to the literature investigating experimentally if and how information about actual inequality (and political position) affects policy preferences. Overall, this paper investigates a relevant question and I enjoyed reading it.

Despite that, I am not sure about the results. I have a number of concerns and I turn to discuss each of these issues in the remainder of this report.

First, the sample is heavily biased to left position and is far to be representative of the overall population. Although, I appreciate that authors are well aware about this limitation, I still think that this is a big issue. The worry is that one would expect the left-wing participants to possibly have quite different preferences from the general population. How much we can take from this work and extend to other groups or even other contexts? Moreover, we have also to take in consideration the timing of the survey. Indeed, voters usually receive more information in the period of elections and may react to them more strongly - at least in the short term.

Second, there are not enough information about groups characteristics. Are participants in the overachiever or underachiever groups different in their characteristics? Did the exclusions of some observations affect the balancing properties of the sample? How much people voting Hamon are different from those voting Macron, for example? These are very important information to understand the validity of the final results.

Third, the findings may be biased by the fact that tasks are not incentivized as they would be in an ideal scenario. In particular, I worry that the lack of impact on most of the voters might be the consequence of the fact that the questions are asked hypothetically. A final question is raised as to whether one can believe that participants might have a desire to please who developed the study. However, there are not enough information on the structure of the survey, so it is very difficult to evaluate the validity of the research design.

6. PLOS authors have the option to publish the peer review history of their article (what does this mean?). If published, this will include your full peer review and any attached files.

Reviewer #1: No

---

## [Author Response · Author response to Decision Letter 0]

4 Nov 2019

Editor’s letter :

I have now collected one review from one expert in the field. Unfortunately, I was unable to find a second reviewer. However, since this review is very detailed and thorough, I have opted for making a decision on your manuscript with only one review. As you will see, the reviewer is quite positive, but has several major comments. Therefore, I would like to invite you to revise your manuscript according to the reviewer's comment. Additionally, after reading your manuscript, I would like to add one more comment. I have myself done a lot of work on redistributive behavior, which I think is quite relevant to your discussion regarding the determinants of redistributive behavior, so you might want to have a look at it (of course, citing this work is not a requirement). In Capraro & Rand (2018) and Tappin & Capraro (2018) we found that redistributive behavior in the trade-off game is driven by moral preferences for doing the right thing and, consequently, it very much depends on how the decision problem is framed; in Capraro (2019), I found that women are more likely than men to choose an equitable distribution over an efficient one. 

Looking forward for the revision. 

References 

Capraro V (2019) Gender differences in the equity-efficiency trade-off. Available at https://ssrn.com/abstract=3386124. 

Capraro V, Rand DG (2018) Do the right thing: Experimental evidence that preferences for moral behavior, rather than equity and efficiency per se, drive human prosociality. Judgment and Decision Making 13, 99-111. 

Tappin BM, Capraro V (2018) Doing good vs. avoiding bad in prosocial choice: A refined test and extension of the morality preference hypothesis. Journal of Experimental Social Psychology 79, 64-70. 

We thank you for bringing this work to our attention. We now refer to Capraro & Rand (2018) in our introduction. In addition, we now include gender as a covariate in our regression analysis. We have found indeed that women redistributed more than men, but this did not change our main results. Regarding this gender effect, we refer to Capraro (2019).

We now report our regression analysis as follows:

“Noteworthy, to evaluate whether our results were robust to changes in model specification, we conducted a new regression analysis, in which we added gender, age and the 5 self-report measures as covariates. This regression revealed that fatalism, views on income equality, and economic patriotism were significant predictors of redistributive behaviors. In addition, we found that gender affected redistribution, with women redistributing more than men, replicating previous findings (e.g. Capraro, 2019; Corneo & Grüner, 2002). We note that the main effect of the First-round vote did not reach significance in this new analysis, but critically the interaction between Status and First-round vote remained significant (Table 2). When examining the effect of Status separately for the 3 groups of voters, again adding the 5 self-report measures as covariates, we found that redistributive behavior was affected by Status only for Mélenchon voters (F(1, 210) = 6.03, p = .0149, ηp 2 = .0279), but not for Hamon voters (F(1, 112) = 2.78, p = .098, ηp 2 = .0242) or Macron voters (F(1, 157) = .790, p = .376, ηp 2 = .0050). replicating our main finding.”

We have also modified Table 2 accordingly. The new Table 2 is as follows:

ANOVA table for redistributive behavior in the disinterested dictator game. The different factors included in the model are the effects of Gender, Age, self-report measures (attitude towards Income inequality, Economic patriotism, Fatalism, Attitude towards France, Political position), Status in the experiment (Overachiever vs. Underachiever), First-round vote and the interaction between Status and First-round vote.

 ηp 2 S.S. d.f. F p 

Gender 0.0097 628 1 4.81 .029 *

Age 0.0006 36 1 0.28 .598 

Income inequality 0.0135 879 1 6.73 .010 **

Economic patriotism 0.0139 906 1 6.94 .009 **

Fatalism 0.0482 3260 1 24.97 <.001 ***

Attitude France 0.0030 193 1 1.48 .225 

Political position 0.0018 119 1 0.91 .340 

Status 0.0029 187 1 1.43 .232 

Vote1 0.0066 425 2 1.63 .197 

Status:Vote1 0.0148 967 2 3.70 .025 *

Residuals 64360 493 

Reviewer #1: Referee’s report - Manuscript ID PONE-D-19-26429 

This paper uses experimental data to examine how redistributive preferences are shaped by the experience of success and political views. This experiment was implemented during the recent presidential election in France. It finds that redistributive behavior is affected primarily by political opinions e.g. Mélenchon voters redistribute more than other groups. In addition, the paper shows that redistributive behavior is influenced by the exogenous manipulation of experiences of success. Interestingly, this effect is not homogenous across political groups and only the Mélenchon voters were significantly affected by the status. 

The paper is well-written and competently conducted. The literature review is fine although I would suggest referring also to the literature investigating experimentally if and how information about actual inequality (and political position) affects policy preferences. Overall, this paper investigates a relevant question and I enjoyed reading it. 

We thank the reviewer for these positive comments. 

Despite that, I am not sure about the results. I have a number of concerns and I turn to discuss each of these issues in the remainder of this report. 

First, the sample is heavily biased to left position and is far to be representative of the overall population. Although, I appreciate that authors are well aware about this limitation, I still think that this is a big issue. The worry is that one would expect the left-wing participants to possibly have quite different preferences from the general population. How much we can take from this work and extend to other groups or even other contexts? Moreover, we have also to take in consideration the timing of the survey. Indeed, voters usually receive more information in the period of elections and may react to them more strongly - at least in the short term. 

We are indeed aware of this limitation, which was already acknowledged in the discussion. In the revised manuscript, we have made explicit the possibility that right-wing or far-right voters could have exhibited a different behavior. 

We now also highlight that the timing of the survey is an important part of the specific context of our study. Generalization to other populations and other contexts is always an issue in experimental studies. However, we believe that since it is an empirical issue, it can be tackled in future work. We have added the following paragraph in the discussion section to address this point:

“In addition, one could argue that a second limitation of the present work is related to the specific timing of the study, which took place during the French presidential election. This specific timing was chosen on purpose for two reasons. One reason was to benefit from the increased interest towards political topics at this time. The other reason was to probe voters’ redistributive behavior at a time that constitutes an important step in the democratic process. However, we acknowledge that it is possible that voters’ behavior in our study is unusual, because of this unusual timing. As voters usually receive more information in the context of an election, it is possible that their reaction to the information we delivered during the study (i.e. the random allocation to the overachiever vs. underachiever group) was affected in these contexts. Whether our results would generalize to another context unrelated to a particular election thus remains an open empirical issue.”

Second, there are not enough information about groups characteristics. Are participants in the overachiever or underachiever groups different in their characteristics? Did the exclusions of some observations affect the balancing properties of the sample? How much people voting Hamon are different from those voting Macron, for example? These are very important information to understand the validity of the final results. 

We thank the reviewer for this comment. We have clarified in the discussion that group characteristics did not differ between Mélenchon voters and Hamon voters (Macron voters were older and more male on average), and between overachievers and underachievers.

“Explanations of this finding in terms of age, sex, or socio-economic status are unlikely in our dataset as Mélenchon voters and Hamon voters did not differ significantly on these variables. In addition, we verified that our Status manipulation was truly random with respect to age, sex, or socio-economic status, which did not differ between overachievers and underachievers.”

In addition, we have now added Table 3 to describe these variables for the 3 group of voters. 

Table 3. Socio-demographic characteristics of participants reporting voting for Mélenchon, Hamon or Macron (in the first round of the election) in our dataset.

First-round vote Age (SD) Gender

(% women) Occupation

(% White Collar) Occupation

(% Student)

Mélenchon 31.21 (12.42) 0.62 0.29 0.45

Hamon 32.57 (14.59) 0.67 0.33 0.45

Macron 37.22 (16.75) 0.45 0.47 0.32

Third, the findings may be biased by the fact that tasks are not incentivized as they would be in an ideal scenario. In particular, I worry that the lack of impact on most of the voters might be the consequence of the fact that the questions are asked hypothetically. A final question is raised as to whether one can believe that participants might have a desire to please who developed the study. However, there are not enough information on the structure of the survey, so it is very difficult to evaluate the validity of the research design. 

Indeed, our measures are based on choices that were not incentivized. We have now added one paragraph in the discussion to highlight this difference between our study and the study of Deffains et al., and the role it may have played in our results.

“Before discussing further our work, we would like to highlight one difference between our study and that of Deffains and colleagues that might be key in explaining this partial discrepancy. In Deffains et al., participants’ redistribution choices in the dictator game had real consequences on the payoffs of other players, whereas in our paradigm redistribution choices were only hypothetical. It has been proposed (e.g. Camerer & Hogarth, 1999) that in the absence of incentives, participants might try to please the experimenter or conform to some social norms, e.g. by being generous in dictator games. It is not clear to us why this “desirability bias” would lead to the specific interaction between Status and First-round vote. Besides, since our experiment was conducted online and responses were anonymous, such an explanation in terms of desirability bias seems unlikely to us. Nevertheless, it is possible that participants who reported voting for Hamon or Macron are more sensitive to the presence of real life incentives, and that a true implementation of their redistribution choices was a necessary feature to obtain the effect of Status. By contrast, it could be that Mélenchon voters are less sensitive to the presence of real incentives, and would exhibit the effect of Status even in the absence of incentives. To evaluate these possibilities, further research would need to compare redistribution choices with and without incentives, for the different groups of voters.

---

## [Editor Report · Decision Letter 1]

14 Nov 2019

PONE-D-19-26429R1

How symbolic success affects redistributive behavior? An experiment based on the 2017 French presidential election

PLOS ONE

Dear Dr. Berthet,

Thanks for your email. Please resubmit the paper by uploading the correct files.

We would appreciate receiving your revised manuscript by Dec 29 2019 11:59PM. To enhance the reproducibility of your results, we recommend that if applicable you deposit your laboratory protocols in protocols.io, where a protocol can be assigned its own identifier (DOI) such that it can be cited independently in the future. For instructions see: http://journals.plos.org/plosone/s/submission-guidelines#loc-laboratory-protocols

We look forward to receiving your revised manuscript.

Kind regards,

Valerio Capraro

Academic Editor

PLOS ONE

Additional Editor Comments (if provided):

Thanks for your email. Please resubmitted the paper attaching the correct files.

---

## [Author Response · Author response to Decision Letter 1]

14 Nov 2019

Editor’s letter :

I have now collected one review from one expert in the field. Unfortunately, I was unable to find a second reviewer. However, since this review is very detailed and thorough, I have opted for making a decision on your manuscript with only one review. As you will see, the reviewer is quite positive, but has several major comments. Therefore, I would like to invite you to revise your manuscript according to the reviewer's comment. Additionally, after reading your manuscript, I would like to add one more comment. I have myself done a lot of work on redistributive behavior, which I think is quite relevant to your discussion regarding the determinants of redistributive behavior, so you might want to have a look at it (of course, citing this work is not a requirement). In Capraro & Rand (2018) and Tappin & Capraro (2018) we found that redistributive behavior in the trade-off game is driven by moral preferences for doing the right thing and, consequently, it very much depends on how the decision problem is framed; in Capraro (2019), I found that women are more likely than men to choose an equitable distribution over an efficient one. 

Looking forward for the revision. 

References 

Capraro V (2019) Gender differences in the equity-efficiency trade-off. Available at https://ssrn.com/abstract=3386124. 

Capraro V, Rand DG (2018) Do the right thing: Experimental evidence that preferences for moral behavior, rather than equity and efficiency per se, drive human prosociality. Judgment and Decision Making 13, 99-111. 

Tappin BM, Capraro V (2018) Doing good vs. avoiding bad in prosocial choice: A refined test and extension of the morality preference hypothesis. Journal of Experimental Social Psychology 79, 64-70. 

We thank you for bringing this work to our attention. We now refer to Capraro & Rand (2018) in our introduction. In addition, we now include gender as a covariate in our regression analysis. We have found indeed that women redistributed more than men, but this did not change our main results. Regarding this gender effect, we refer to Capraro (2019).

We now report our regression analysis as follows:

“Noteworthy, to evaluate whether our results were robust to changes in model specification, we conducted a new regression analysis, in which we added gender, age and the 5 self-report measures as covariates. This regression revealed that fatalism, views on income equality, and economic patriotism were significant predictors of redistributive behaviors. In addition, we found that gender affected redistribution, with women redistributing more than men, replicating previous findings (e.g. Capraro, 2019; Corneo & Grüner, 2002). We note that the main effect of the First-round vote did not reach significance in this new analysis, but critically the interaction between Status and First-round vote remained significant (Table 2). When examining the effect of Status separately for the 3 groups of voters, again adding the 5 self-report measures as covariates, we found that redistributive behavior was affected by Status only for Mélenchon voters (F(1, 210) = 6.03, p = .0149, ηp 2 = .0279), but not for Hamon voters (F(1, 112) = 2.78, p = .098, ηp 2 = .0242) or Macron voters (F(1, 157) = .790, p = .376, ηp 2 = .0050). replicating our main finding.”

We have also modified Table 2 accordingly. The new Table 2 is as follows:

ANOVA table for redistributive behavior in the disinterested dictator game. The different factors included in the model are the effects of Gender, Age, self-report measures (attitude towards Income inequality, Economic patriotism, Fatalism, Attitude towards France, Political position), Status in the experiment (Overachiever vs. Underachiever), First-round vote and the interaction between Status and First-round vote.

 ηp 2 S.S. d.f. F p 

Gender 0.0097 628 1 4.81 .029 *

Age 0.0006 36 1 0.28 .598 

Income inequality 0.0135 879 1 6.73 .010 **

Economic patriotism 0.0139 906 1 6.94 .009 **

Fatalism 0.0482 3260 1 24.97 ***

Attitude France 0.0030 193 1 1.48 .225 

Political position 0.0018 119 1 0.91 .340 

Status 0.0029 187 1 1.43 .232 

Vote1 0.0066 425 2 1.63 .197 

Status:Vote1 0.0148 967 2 3.70 .025 *

Residuals 64360 493 

Reviewer #1: Referee’s report - Manuscript ID PONE-D-19-26429 

This paper uses experimental data to examine how redistributive preferences are shaped by the experience of success and political views. This experiment was implemented during the recent presidential election in France. It finds that redistributive behavior is affected primarily by political opinions e.g. Mélenchon voters redistribute more than other groups. In addition, the paper shows that redistributive behavior is influenced by the exogenous manipulation of experiences of success. Interestingly, this effect is not homogenous across political groups and only the Mélenchon voters were significantly affected by the status. 

The paper is well-written and competently conducted. The literature review is fine although I would suggest referring also to the literature investigating experimentally if and how information about actual inequality (and political position) affects policy preferences. Overall, this paper investigates a relevant question and I enjoyed reading it. 

We thank the reviewer for these positive comments. 

Regarding the comment about the experimental literature, we have now expended the third paragraph of introduction to include some references showing that providing information about actual inequality affects support for redistribution. The paragraph now reads as follows:

“Surveys have shown that such beliefs about the determinants of inequality are not homogeneous across the population (e.g. Cozzarelli, Wilkinson, Tagler, 2001). Relatedly, support for redistributive policies varies across social groups defined by race, gender, age or socioeconomic status (Keely & Tan, 2008). In the United States, whites are more averse to redistribution than blacks, even after controlling for individual characteristics such as income, education, etc. (e.g., Gilens, 1999; Alesina & La Ferrara, 2005). Past upward mobility also decreases the support for redistribution (e.g. Guillaud & Sauger, 2013; Alesina & La Ferrara, 2005). Some of these observations have been confirmed by experimental data. For instance, when participants are presented with mock news articles reporting high (vs. low) rates of social mobility, their tolerance for inequality increased (Shaffir, Wiwad, Aknin, 2016). Providing American adults with factual information about the rise of inequalities in the United States (vs. control information) increased their beliefs that economic inequalities are due to structural rather than individual factors and increased support for redistribution (McCall et al., 2017; Boudreau & MacKenzie, 2018).”

Despite that, I am not sure about the results. I have a number of concerns and I turn to discuss each of these issues in the remainder of this report. 

First, the sample is heavily biased to left position and is far to be representative of the overall population. Although, I appreciate that authors are well aware about this limitation, I still think that this is a big issue. The worry is that one would expect the left-wing participants to possibly have quite different preferences from the general population. How much we can take from this work and extend to other groups or even other contexts? Moreover, we have also to take in consideration the timing of the survey. Indeed, voters usually receive more information in the period of elections and may react to them more strongly - at least in the short term. 

We are indeed aware of this limitation due to the bias in our sample, which was already acknowledged in the discussion. In the revised manuscript, we have made more explicit the possibility that right-wing or far-right voters could have exhibited a different behavior. 

We now also highlight that the timing of the survey is an important part of the specific context of our study. Generalization to other populations and other contexts is always an issue in experimental studies. However, we believe that since it is an empirical issue, it can be tackled in future work. We have added the following paragraph in the discussion section to address this point:

“In addition, one could argue that a second limitation of the present work is related to the specific timing of the study, which took place during the French presidential election. This specific timing was chosen on purpose for two reasons. One reason was to benefit from the increased interest towards political topics at this time. The other reason was to probe voters’ redistributive behavior at a time that constitutes an important step in the democratic process. However, we acknowledge that it is possible that voters’ behavior in our study is unusual, because of this unusual timing. Voters may receive more information in the context of an election, and they may react more strongly to information delivered in this context. Whether our results would generalize to another context unrelated to a particular election thus remains an open empirical issue.”

Second, there are not enough information about groups characteristics. Are participants in the overachiever or underachiever groups different in their characteristics? Did the exclusions of some observations affect the balancing properties of the sample? How much people voting Hamon are different from those voting Macron, for example? These are very important information to understand the validity of the final results. 

We thank the reviewer for this comment. We have clarified in the discussion that group characteristics did not differ between Mélenchon voters and Hamon voters (Macron voters were older and more male on average), and between overachievers and underachievers.

“Explanations of this finding in terms of age, sex, or socio-economic status are unlikely in our dataset as Mélenchon voters and Hamon voters did not differ significantly on these variables (Table 3). In addition, we verified that our Status manipulation was truly random with respect to age, sex, or socio-economic status, which did not differ between overachievers and underachievers.”

In addition, we have now added Table 3 to describe these variables for the 3 group of voters. 

Table 3. Socio-demographic characteristics of participants reporting voting for Mélenchon, Hamon or Macron (in the first round of the election) in our dataset.

First-round vote Age (SD) Gender

(% women) Occupation

(% White Collar) Occupation

(% Student)

Mélenchon 31.21 (12.42) 0.62 0.29 0.45

Hamon 32.57 (14.59) 0.67 0.33 0.45

Macron 37.22 (16.75) 0.45 0.47 0.32

Third, the findings may be biased by the fact that tasks are not incentivized as they would be in an ideal scenario. In particular, I worry that the lack of impact on most of the voters might be the consequence of the fact that the questions are asked hypothetically. A final question is raised as to whether one can believe that participants might have a desire to please who developed the study. However, there are not enough information on the structure of the survey, so it is very difficult to evaluate the validity of the research design. 

Indeed, our measures are based on choices that were not incentivized. We have now added one paragraph in the discussion to highlight this difference between our study and the study of Deffains et al., and the role it may have played in our results.

“Our main result is that redistributive behavior is influenced by the exogenous manipulation of Status only in a subgroup of participants, specifically those who reported voting for Mélenchon. Therefore, our study partially replicated the findings of Deffains and colleagues (2016). This partial discrepancy between our study and that of Deffains might be due to incentives. In Deffains’ study, participants’ redistribution choices in the dictator game had real consequences on the payoffs of other players, whereas in our paradigm redistribution choices were only hypothetical. It has been proposed (e.g. Camerer & Hogarth, 1999) that in the absence of incentives, participants might try to please the experimenter or conform to some social norms, e.g. by being generous in dictator games. However, in our experiment, participants redistributed less than in Deffains’ study: our mean allocation to A was 60.08 while the corresponding value in Deffains’ study would be 57.56. Besides, it is not clear to us why this desirability bias would lead to the specific interaction between Status and First-round vote. Furthermore, since our experiment was conducted online and responses were anonymous, such an explanation in terms of desirability bias seems unlikely to us. Nevertheless, it is possible that incentives might have influenced our results independently of the desirability bias. Participants who reported voting for Hamon or Macron might be more sensitive to the presence of real life incentives than Mélenchon voters. Thus, incentivizing redistribution choices might be a necessary feature to obtain the effect of Status in Hamon or Macron voters, whereas Mélenchon voters would exhibit the effect of Status even in the absence of incentives. To evaluate these possibilities, further research would need to compare redistribution choices with and without incentives, for the different groups of voters.”

---

## [Decision Letter · Decision Letter 2]

6 Dec 2019

PONE-D-19-26429R2

How symbolic success affects redistributive behavior? An experiment based on the 2017 French presidential election

PLOS ONE

Dear Dr. Berthet,

Thank you for submitting your manuscript to PLOS ONE. After careful consideration, we feel that it has merit but does not fully meet PLOS ONE’s publication criteria as it currently stands. Therefore, we invite you to submit a revised version of the manuscript that addresses the points raised during the review process.

We would appreciate receiving your revised manuscript by Jan 20 2020 11:59PM. To enhance the reproducibility of your results, we recommend that if applicable you deposit your laboratory protocols in protocols.io, where a protocol can be assigned its own identifier (DOI) such that it can be cited independently in the future. For instructions see: http://journals.plos.org/plosone/s/submission-guidelines#loc-laboratory-protocols

We look forward to receiving your revised manuscript.

Kind regards,

Valerio Capraro

Academic Editor

PLOS ONE

Additional Editor Comments (if provided):

The referee still has some major comments. Please do your best to address their comments.

Reviewers' comments:

Reviewer's Responses to Questions

**Comments to the Author**

1. If the authors have adequately addressed your comments raised in a previous round of review and you feel that this manuscript is now acceptable for publication, you may indicate that here to bypass the “Comments to the Author” section, enter your conflict of interest statement in the “Confidential to Editor” section, and submit your "Accept" recommendation.

Reviewer #1: (No Response)

2. Is the manuscript technically sound, and do the data support the conclusions?

Reviewer #1: Partly

3. Has the statistical analysis been performed appropriately and rigorously? 

Reviewer #1: No

4. Have the authors made all data underlying the findings in their manuscript fully available?

Reviewer #1: (No Response)

5. Is the manuscript presented in an intelligible fashion and written in standard English?

Reviewer #1: Yes

6. Review Comments to the Author

Reviewer #1: Referee’s report - Manuscript ID PONE-D-19-26429R2

Many thanks for the answers. However, I think that there is still something in need of clarification. In particular, my main concerns are about the exclusions of some observations and the balancing property of the sample. Please find below my comments to the previous answers.

--

REFEREE: First, the sample is heavily biased to left position and is far to be representative of the overall population. Although, I appreciate that authors are well aware about this limitation, I still think that this is a big issue. The worry is that one would expect the left-wing participants to possibly have quite different preferences from the general population. How much we can take from this work and extend to other groups or even other contexts?

AUTHOR/S: We are indeed aware of this limitation due to the bias in our sample, which was already acknowledged in the discussion. In the revised manuscript, we have made more explicit the possibility that right-wing or far-right voters could have exhibited a different behavior. Generalization to other populations and other contexts is always an issue in experimental studies. However, we believe that since it is an empirical issue, it can be tackled in future work.

REFEREE R2: Many thanks for your response. However, I`m still not sure about this point. The current title is a bit misleading. I would suggest you to make more explicit that your analysis focuses on left-wing voters. This also requires some additional effort in terms of framing the discussion. Sometimes, it seems that you are referring to the general population.

I also agree that representativity is always an issue in experimental studies. But, we run experiment to understand something more about preferences, behaviors etc. My main concern with this study refers to the fact that the sample is totally biased to the left and in particular to the “French left” during the time of the 2016 election … So, what can I learn from it? What in terms of value added to the existing literature? This something that should be explained better in the paper.

REFEREE: Second, there are not enough information about groups characteristics. Are participants in the overachiever or underachiever groups different in their characteristics? Did the exclusions of some observations affect the balancing properties of the sample? How much

people voting Hamon are different from those voting Macron, for example? These are very important information to understand the validity of the final results.

AUTHOR/S: We thank the reviewer for this comment. We have clarified in the discussion that group characteristics did not differ between Mélenchon voters and Hamon voters (Macron voters were older and more male on average), and between overachievers and underachievers.

REFEREE R2: I think that some answers are missing. I would suggest to address these points in the next version of the paper. Was the randomization before or after the exclusion of the observations? In my understanding, the randomization was before that some observations were excluded. Did the exclusion of these observations affect the balancing properties of the sample? It is always not good to exclude observations. If you do that, you should show that this will not affect your final results. My main concern is still: Are participants in the overachiever or underachiever groups different in their characteristics? author/s should show a table with these statistics and not only reporting two lines saying … “In addition, we verified that our Status manipulation was truly random with respect to age, sex, or socio-economic status, which did not differ between overachievers and underachievers.”

REFEREE: Third, the findings may be biased by the fact that tasks are not incentivized as they would be in an ideal scenario. In particular, I worry that the lack of impact on most of the voters might be the consequence of the fact that the questions are asked hypothetically. A final question is raised as to whether one can believe that participants might have a desire to please who developed the study. However, there are not enough information on the structure of the survey, so it is very difficult to evaluate the validity of the research design.

AUTHOR/S: Indeed, our measures are based on choices that were not incentivized. We have now added one paragraph in the discussion to highlight this difference between our study and the study of Deffains et al., and the role it may have played in our results.

REFEREE R2: I`m fine with the discussion on the incentives but, again, the author/s did not reply to my final comment (or provide more information) about the potential desire of participants to please who developed the study.

7. PLOS authors have the option to publish the peer review history of their article (what does this mean?). If published, this will include your full peer review and any attached files.

Reviewer #1: No

---

## [Author Response · Author response to Decision Letter 2]

10 Dec 2019

Reviewer #1: Referee’s report - Manuscript ID PONE-D-19-26429R2 

Many thanks for the answers. However, I think that there is still something in need of clarification. In particular, my main concerns are about the exclusions of some observations and the balancing property of the sample. Please find below my comments to the previous answers. 

REFEREE: First, the sample is heavily biased to left position and is far to be representative of the overall population. Although, I appreciate that authors are well aware about this limitation, I still think that this is a big issue. The worry is that one would expect the left-wing participants to possibly have quite different preferences from the general population. How much we can take from this work and extend to other groups or even other contexts? 

AUTHOR/S: We are indeed aware of this limitation due to the bias in our sample, which was already acknowledged in the discussion. In the revised manuscript, we have made more explicit the possibility that right-wing or far-right voters could have exhibited a different behavior. Generalization to other populations and other contexts is always an issue in experimental studies. However, we believe that since it is an empirical issue, it can be tackled in future work. 

REFEREE R2: Many thanks for your response. However, I`m still not sure about this point. The current title is a bit misleading. I would suggest you to make more explicit that your analysis focuses on left-wing voters. This also requires some additional effort in terms of framing the discussion. Sometimes, it seems that you are referring to the general population. 

I also agree that representativity is always an issue in experimental studies. But, we run experiment to understand something more about preferences, behaviors etc. My main concern with this study refers to the fact that the sample is totally biased to the left and in particular to the “French left” during the time of the 2016 election … So, what can I learn from it? What in terms of value added to the existing literature? This something that should be explained better in the paper. 

AUTHOR/S R2: We changed the title, and made our focus on left-wing voters more explicit in the abstract and in the main text.

In terms of representativity / general interest of our results, we now mention in the discussion that: 

“We note that although our results may not be representative of right-wing voters, one could envision that they would generalize for left-wing voters in some other countries. Indeed, in the last decade, western democracies have seen a polarization of opinions, with a crisis of the traditional parties and a rise of support for extreme populist parties. Examples of populist far-left parties are die Linke in Germany, Podemos in Spain, Siriza in Greece, La France Insoumise (Jean-Luc Mélenchon’s party) in France. According to Rooduijn and Akkerman (2017) these radical left parties have in common that “They do not focus on the ‘proletariat’, but glorify a more general category: the ‘good people’” , contrary to former communist parties and that ”they do not reject the system of liberal democracy as such, but only criticize the political and/or economic elites within that system”. Our results regarding the susceptibility of Mélenchon voters to status manipulation could thus be evaluated and replicated in other countries.”

Regarding how our study contributes to the existing literature, we now highlight that our general finding that Mélenchon voters displayed a different behavior than Hamon and Macron voters connects with recent findings showing that supporters of extreme political groups have different characteristics from those with more moderate views (e.g. Rooduijn, 2018; Hanel et al., 2019). We have re-written the last paragraph in the discussion section to address this point:

“To conclude, our findings revealed that self-reported far-left voters turned out to be the more sensitive to the exogenous manipulation of symbolic success. This leads to three remarks. Firstly, we need further research to better understand to what extent, and in which groups, redistributive behavior can be manipulated through exogenous manipulations of experience of success. In particular, further studies are needed that shall use a proper manipulation of symbolic success and representative samples in terms of political and socio-economic features. Secondly, our findings suggest that the various political groups process information differently, that is, they are not cognitively homogeneous (e.g., Amodio, Jost, Master, Yee, 2007; Rollwage, Dolan, & Fleming, 2018; Clarkson et al., 2015). Finally, and more broadly, the fact that Mélenchon voters displayed a different behavior than Hamon and Macron voters extends recent findings showing that supporters of extreme political groups have different characteristics from those with more moderate views, although they are not necessarily different on socio-demographic variables such as age or level of education (e.g. Rooduijn, 2018). For instance, Hanel, Zarzeczna, and Haddock (2019) reported that extreme (left-wing or right-wing) supporters are usually more heterogeneous than moderate ones in terms of human values and politics-related variables such as attitudes toward immigrants and trust in institutions. In the current social and political context, we believe that understanding further these differences, especially whether some groups are more susceptible to influence than others, appears a worthwhile subject for future research. Using controlled experiments during political elections can be a useful tool in such research.”

The added references are:

Rooduijn, M., & Akkerman, T. (2017). Flank attacks: Populism and left-right radicalism in Western Europe. Party Politics, 23(3), 193-204.

Rooduijn, M. (2018). What unites the voter bases of populist parties? Comparing the electorates of 15 populist parties. European Political Science Review, 10(3), 351-368.

Hanel, P. H. P., Zarzeczna, N., & Haddock, G. (2019). Sharing the same political ideology yet endorsing different values: left- and right-wing political supporters are more heterogeneous than moderates. Social Psychological and Personality Science, 10, 874-882.

REFEREE: Second, there are not enough information about groups characteristics. Are participants in the overachiever or underachiever groups different in their characteristics? Did the exclusions of some observations affect the balancing properties of the sample? How much people voting Hamon are different from those voting Macron, for example? These are very important information to understand the validity of the final results. 

AUTHOR/S: We thank the reviewer for this comment. We have clarified in the discussion that group characteristics did not differ between Mélenchon voters and Hamon voters (Macron voters were older and more male on average), and between overachievers and underachievers. 

REFEREE R2: I think that some answers are missing. I would suggest to address these points in the next version of the paper. Was the randomization before or after the exclusion of the observations? In my understanding, the randomization was before that some observations were excluded. Did the exclusion of these observations affect the balancing properties of the sample? It is always not good to exclude observations. If you do that, you should show that this will not affect your final results. My main concern is still: Are participants in the overachiever or underachiever groups different in their characteristics? author/s should show a table with these statistics and not only reporting two lines saying … “In addition, we verified that our Status manipulation was truly random with respect to age, sex, or socio-economic status, which did not differ between overachievers and underachievers.” 

AUTHOR/S R2: We have updated Table 3 and added Table 4 to provide more information about our groups as well as the statistics for the comparison between overachiever and underachiever. We have moved these two tables up in the main text so they are now Table 1 and Table 2.

Table 1

Socio-demographic characteristics of participants reporting voting for Mélenchon, Hamon or Macron in the first round of the election in our dataset. Participants who reported another vote are pooled together in this table, and were not analyzed further in the present study.

First-round vote N Age (SD) Gender

(% women) Occupation

(% White Collar) Occupation

(% Student) Status

(% overachiever)

Mélenchon 219 31.21 (12.42) 0.62 0.29 0.45 0.46

Hamon 121 32.57 (14.59) 0.67 0.33 0.45 0.57

Macron 166 37.22 (16.75) 0.45 0.47 0.32 0.47

Other 120 32.40 (14.65) 0.46 0.23 0.46 0.43

Table 2 

Comparison of the socio-demographic characteristics of participants in the overachiever and underachiever conditions in our final sample (i.e. including only participants who reported voting for Macron, Hamon or Mélenchon).

Condition N Age (SD) Gender

(% women) Occupation

(% White Collar) Occupation

(% Student)

Overachiever 248 33.73 (15.34) 0.54 0.38 0.41

Underachiever 258 33.28 (14.06) 0.60 0.34 0.40

T 0.344 -1.371 0.888 0.187

p 0.731 0.171 0.375 0.852

REFEREE: Third, the findings may be biased by the fact that tasks are not incentivized as they would be in an ideal scenario. In particular, I worry that the lack of impact on most of the voters might be the consequence of the fact that the questions are asked hypothetically. A final question is raised as to whether one can believe that participants might have a desire to please who developed the study. However, there are not enough information on the structure of the survey, so it is very difficult to evaluate the validity of the research design. 

AUTHOR/S: Indeed, our measures are based on choices that were not incentivized. We have now added one paragraph in the discussion to highlight this difference between our study and the study of Deffains et al., and the role it may have played in our results. 

REFEREE R2: I`m fine with the discussion on the incentives but, again, the author/s did not reply to my final comment (or provide more information) about the potential desire of participants to please who developed the study. 

AUTHOR/S R2: Our paragraph also addressed the issue of a potential desirability bias. Specifically, in our previous revision we wrote: “It has been proposed (e.g. Camerer & Hogarth, 1999) that in the absence of incentives, participants might try to please the experimenter or conform to some social norms, e.g. by being generous in dictator games. However, in our experiment, participants redistributed less than in Deffains’ study: our mean allocation to A was 60.08 while the corresponding value in Deffains’ study would be 57.56. Besides, it is not clear to us why this desirability bias would lead to the specific interaction between Status and First-round vote. Furthermore, since our experiment was conducted online and responses were anonymous, such an explanation in terms of desirability bias seems unlikely to us.”

We have now modified this passage to clarify this and better separate the discussion of incentives and the discussion of desirability bias. We now write in the discussion:

“It has been proposed (e.g. Camerer & Hogarth, 1999) that in the absence of incentives, participants might try to please the experimenter or conform to some social norms, e.g. by being generous in dictator games. Could this desirability bias explain our results or the difference between our study and Deffains’ study? We believe that such an explanation is unlikely for several reasons. First, if a desirability bias was more present our study than in Deffains’ study, then we should have observed more redistribution in our participants. However, in our experiment, participants redistributed less than in Deffains’ study: our mean allocation to A was 60.08 while the corresponding value in Deffains’ study would be 57.56. Second, and more generally, it is not clear to us why this desirability bias would lead to the specific interaction between Status and First-round vote. Third, the instructions given to participants (see Appendix A) did not refer to the aim of our experiment, so participants were naïve about our hypothesis. Had they tried to guess our expectations, we would have found an effect of status on fatalism, which we did not observe either in the full sample (p=.52) nor in Mélenchon voters (p=.35), whose redistributive behavior was affected by status however. Finally, our experiment was conducted online and responses were anonymous, so participants have no pressure to please the experimenter or conform to social norms.”

---

## [Decision Letter · Decision Letter 3]

3 Jan 2020

PONE-D-19-26429R3

How does symbolic success affect redistribution in left-wing voters? A focus on the 2017 French presidential election

PLOS ONE

Dear Dr. Berthet,

Thank you for submitting your manuscript to PLOS ONE. After careful consideration, we feel that it has merit but does not fully meet PLOS ONE’s publication criteria as it currently stands. Therefore, we invite you to submit a revised version of the manuscript that addresses the points raised during the review process.

We would appreciate receiving your revised manuscript by Feb 17 2020 11:59PM. To enhance the reproducibility of your results, we recommend that if applicable you deposit your laboratory protocols in protocols.io, where a protocol can be assigned its own identifier (DOI) such that it can be cited independently in the future. For instructions see: http://journals.plos.org/plosone/s/submission-guidelines#loc-laboratory-protocols

We look forward to receiving your revised manuscript.

Kind regards,

Valerio Capraro

Academic Editor

PLOS ONE

Additional Editor Comments (if provided):

The reviewer suggests minor revisions. Please address them at your earliest convenience. I am looking forward for the revision.

Reviewers' comments:

Reviewer's Responses to Questions

**Comments to the Author**

1. If the authors have adequately addressed your comments raised in a previous round of review and you feel that this manuscript is now acceptable for publication, you may indicate that here to bypass the “Comments to the Author” section, enter your conflict of interest statement in the “Confidential to Editor” section, and submit your "Accept" recommendation.

Reviewer #1: (No Response)

2. Is the manuscript technically sound, and do the data support the conclusions?

Reviewer #1: Yes

3. Has the statistical analysis been performed appropriately and rigorously? 

Reviewer #1: Yes

4. Have the authors made all data underlying the findings in their manuscript fully available?

Reviewer #1: (No Response)

5. Is the manuscript presented in an intelligible fashion and written in standard English?

Reviewer #1: Yes

6. Review Comments to the Author

Reviewer #1: Referee’s report - Manuscript ID PONE-D-19-26429R3

Many thanks for these answers. As I said in my previous reports, I truly believe that this paper investigates a relevant question and I enjoyed reading it. Over this process, I believe that the paper has improved a lot.

Yet, there is still something in need of clarification.

As I said in my previous email, it is not good to exclude observations. Therefore, I wonder if it is possible to see (and report in the paper – maybe in the annex) the results of the regression including the observations excluded. Also, I would be happy to see if participants in the overachiever or underachiever groups are different not only in their socio – demographic characteristics but also on “Vote 1”.

Reading again the paper, I have also some additional suggestion/recommendation. I think that the paper would improve if all the results discussed in the paper are also summarized using tables. It would be great to have a table summarizing the results discussed between the line 191 and 193. By contrast, I think that the results reported in Table 4 could not be considered valid. Some of the controls might have been affected by the manipulation e.g. fatalism and views on income inequality. Therefore, the inclusion of these variables as controls may introduce some bias in the estimation. In spite of that, I would suggest to replicate the baseline estimation adding only gender and age as controls.

Finally, I think that the paper would benefit for a better explanation of the variable used in the analysis. How is the First-round vote variable used in the regression is defined after the exclusion of some observations?

7. PLOS authors have the option to publish the peer review history of their article (what does this mean?). If published, this will include your full peer review and any attached files.

Reviewer #1: No

---

## [Author Response · Author response to Decision Letter 3]

8 Jan 2020

Reviewer #1: Referee’s report - Manuscript ID PONE-D-19-26429R3

REFEREE R3: Many thanks for these answers. As I said in my previous reports, I truly believe that this paper investigates a relevant question and I enjoyed reading it. Over this process, I believe that the paper has improved a lot. Yet, there is still something in need of clarification.

As I said in my previous email, it is not good to exclude observations. Therefore, I wonder if it is possible to see (and report in the paper – maybe in the annex) the results of the regression including the observations excluded. 

AUTHOR/S R3: Regarding the exclusion of observations, we mentioned that few observations were excluded due to a technical error in the data collection. In the paragraph “Description of our sample”, we added the following sentence:

“(due to a technical error in the data collection, redistribution choices could only be analyzed for 626 participants).”

We ran the regression including all observations and added the results in the Annex (Appendix A). Following the recommendation of the reviewer below, we ran the regression adding only gender and age as covariates. Moreover, as some categories of First-round vote have few observations (e.g. Arthaud, Poutou, Lassalle, Cheminade, Asselineau, Dupont-Aignan), observations were merged as follows: Arthaud and Poutou votes were merged with Mélenchon (N = 235); Lassalle votes were merged with Macron (N = 168); Cheminade votes were merged with Fillon (N = 36); Asselineau, Dupont-Aignan, and Le Pen votes were merged in the “FarRight” category (N = 19); Blank votes and abstentions were merged in the “NoVote” category (N = 47). We added the following sentence in the main text:

“For completeness, we also report in Appendix A the results of a regression over all participants, including those who reported a different First-round vote.”

Appendix A: ANOVA table for redistributive behavior in the disinterested dictator game including all groups of voters.

 ηp 2 S.S. d.f. F p

Gender 0.0113 1074 1 7.006 0.008

Age 0.0039 367 1 2.396 0.122

Status 0.0006 57 1 0.373 0.542

Vote1 0.0683 6878 5 8.971 <0.001

Status:Vote1 0.0138 1313 5 1.713 0.130

Residuals 93848 612 

REFEREE R3: Also, I would be happy to see if participants in the overachiever or underachiever groups are different not only in their socio – demographic characteristics but also on “Vote 1”. 

AUTHOR/S: We verified that Vote 1 and Status were not associated, by running an ANOVA with Status as a dependent variable and Vote 1 as an independent variable (F(2,503)=2.06, p=.13). In addition, we conducted separate tests comparing the proportion of votes for Mélenchon, Hamon and Macron between overachievers and underachievers. These tests indicated no difference between groups. This information about Vote 1 is now included in the table (Table 2). 

Condition N Age (SD) Gender

(% women) First-round vote

Mélenchon / Hamon / Macron

Overachiever 248 33.73 (15.34) 0.54 0.41 / 0.28 / 0.31

Underachiever 258 33.28 (14.06) 0.60 0.46 / 0.20 / 0.34

test t = 0.344 χ^2^ = 1.64 χ^2^ = 1.10 / 3.67 / 0.29

p 0.731 0.2 0.29 / 0.06 / 0.59

REFEREE R3: Reading again the paper, I have also some additional suggestion/recommendation. I think that the paper would improve if all the results discussed in the paper are also summarized using tables. It would be great to have a table summarizing the results discussed between the line 191 and 193. 

AUTHOR/S: We replaced Figure 3 by a new table (Table 4) indicating means and standard errors of the amount of money allocated to A in the Disinterested Dictator Game as a function of First-round vote and Status.

Table 4

Means and standard errors of the amount of money allocated to A in the Disinterested Dictator Game as a function of First-round vote and Status.

First-round vote

Status Mélenchon Hamon Macron

Overachiever 60.5 (1.28) 59.2 (1.30) 64.4 (1.58)

Underachiever 56.7 (0.91) 62.3 (1.86) 63.0 (1.33)

REFEREE R3: By contrast, I think that the results reported in Table 4 could not be considered valid. Some of the controls might have been affected by the manipulation e.g. fatalism and views on income inequality. Therefore, the inclusion of these variables as controls may introduce some bias in the estimation. In spite of that, I would suggest to replicate the baseline estimation adding only gender and age as controls.

AUTHOR/S: We followed the recommendation of the reviewer and we now report the regression analysis adding only gender and age as covariates (Table 5). The results were virtually unchanged. 

Table 5

ANOVA table for redistributive behavior in the disinterested dictator game. The different factors included in the model are the effects of Gender, Age, Status in the experiment (Overachiever vs. Underachiever), First-round vote and the interaction between Status and First-round vote.

 ηp 2 S.S. d.f. F p

Gender 0.0080 584 1 4.001 .046

Age 0.0051 373 1 2.553 .111

Status 0.0026 188 1 1.291 .256

Vote1 0.0239 1781 2 6.101 .002

Status:Vote1 0.0119 874 2 2.992 .051

Residuals 72701 498 

We have re-written the paragraph accordingly:

“Noteworthy, to evaluate whether our results were robust to changes in model specification, we conducted a new regression analysis, in which we added gender and age as covariates (Table 5). This regression revealed that gender affected redistribution, with women redistributing more than men, replicating previous findings (e.g. Capraro, 2019; Corneo & Grüner, 2002). This analysis also indicated a main effect of the First-round vote, and confirmed the interaction between Status and First-round vote. When examining the effect of Status separately for the 3 groups of voters, again adding gender and age as covariates, we found that redistributive behavior was affected by Status only for Mélenchon voters (F(1, 215) = 5.54, p = .020, ηp 2 = .0251), but not for Hamon voters (F(1, 117) = 2.09, p = .151, ηp 2 = .0176) or Macron voters (F(1, 162) = 0.454, p = .502, ηp 2 = .0028) replicating our main finding. For completeness, we also report in Appendix A the results of a regression over all participants, including those who reported a different First-round vote.”

REFEREE R3: Finally, I think that the paper would benefit for a better explanation of the variable used in the analysis. How is the First-round vote variable used in the regression is defined after the exclusion of some observations?

AUTHOR/S: 

In the paragraph “Description of our sample”, we have made more explicit the fact that in our analysis, the categories of the First-round vote variable are restricted to Mélenchon, Hamon, and Macron. We added the following sentence:

“Accordingly, in what follows the variable “First-round vote” is a categorical variable with 3 possible values, namely Mélenchon, Hamon, and Macron.”

---

## [Editor Report · Decision Letter 4]

30 Jan 2020

How does symbolic success affect redistribution in left-wing voters? A focus on the 2017 French presidential election

PONE-D-19-26429R4

Dear Dr. Berthet,

We are pleased to inform you that your manuscript has been judged scientifically suitable for publication and will be formally accepted for publication once it complies with all outstanding technical requirements.

With kind regards,

Valerio Capraro

Academic Editor

PLOS ONE
---

## [Editor Report · Acceptance letter]

6 Feb 2020

PONE-D-19-26429R4 

How does symbolic success affect redistribution in left-wing voters? A focus on the 2017 French presidential election 

Dear Dr. Berthet:

I am pleased to inform you that your manuscript has been deemed suitable for publication in PLOS ONE. Congratulations! Your manuscript is now with our production department. 

With kind regards,

on behalf of

Dr. Valerio Capraro 

Academic Editor

PLOS ONE